# DNAEdit: Direct Noise Alignment for Text-Guided Rectified Flow Editing

**Chenxi Xie**[1,2,*], **Minghan Li**[3,*], **Shuai Li**[1], **Yuhui Wu**[1,2], **Qiaosi Yi**[1,2], **Lei Zhang**[1,2,†]

[1]The Hong Kong Polytechnic University, [2]OPPO Research Institute,
[3]Harvard University

chenxi.xie@connect.polyu.hk  mili4@meei.harvard.edu  cslzhang@comp.polyu.edu.hk
[*]Equal contribution    [†]Corresponding author
Project Page: https://xiechenxi99.github.io/DNAEdit/

## Abstract

Leveraging the powerful generation capability of large-scale pretrained text-to-image models, training-free methods have demonstrated impressive image editing results. Conventional diffusion-based methods, as well as recent rectified flow (RF)-based methods, typically reverse synthesis trajectories by gradually adding noise to clean images, during which the noisy latent at the current timestep is used to approximate that at the next timesteps, introducing accumulated drift and degrading reconstruction accuracy. Considering the fact that in RF the noisy latent is estimated through direct interpolation between Gaussian noises and clean images at each timestep, we propose Direct Noise Alignment (DNA), which directly refines the desired Gaussian noise in the noise domain, significantly reducing the error accumulation in previous methods. Specifically, DNA estimates the velocity field of the interpolated noised latent at each timestep and adjusts the Gaussian noise by computing the difference between the predicted and expected velocity field. We validate the effectiveness of DNA and reveal its relationship with existing RF-based inversion methods. Additionally, we introduce a Mobile Velocity Guidance (MVG) to control the target prompt-guided generation process, balancing image background preservation and target object editability. DNA and MVG collectively constitute our proposed method, namely DNAEdit. Finally, we introduce DNA-Bench, a long-prompt benchmark, to evaluate the performance of advanced image editing models. Experimental results demonstrate that our DNAEdit achieves superior performance to state-of-the-art text-guided editing methods. Our code, model, and benchmark will be made publicly available.

## 1 Introduction

Recent advances in text-to-image (T2I) generation have been driven by Rectified Flow (RF)-based models [12, 11], which significantly reduce sampling timesteps, enabling faster generation. Leveraging large-scale T2I models such as SD3 [3] and FLUX [8], training-free text-guided image editing methods [17, 22, 7, 30] can achieve high-quality editing results with fewer sampling steps. Existing RF-based editing methods [17, 7, 2, 22, 25, 27, 30] typically follow earlier Diffusion Model (DM)-based editing approaches [15, 6, 14], first reversing the generative trajectory by gradually adding noise to the clean image, obtaining an inverted noise and then re-denosing it under new conditions to generate the edited image, as shown by the black path in Fig. 1 (a). The inverted noise is critical for the preservation of fidelity, as it retains the structural information of the reference image, ensuring consistency between the edited and the reference images. However, this inversion process introduces

This work is supported by the PolyU-OPPO Joint Innovative Research Center.

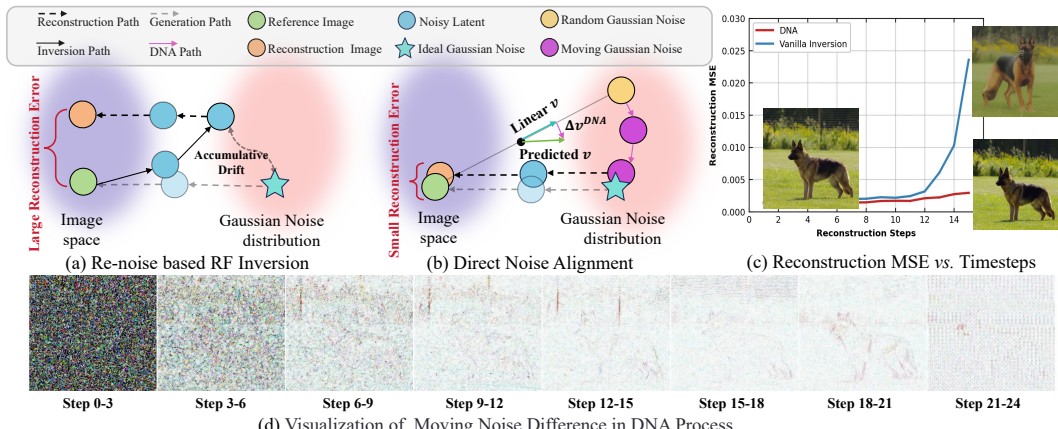

Figure 1: Illustration of (a) existing re-noise based RF inversion methods and (b) our DNA. The ideal Gaussian noise refers to the noise that can exactly reproduce the reference image. (c) The curves of reconstruction MSE *vs.* timesteps for DNA and re-noise based RF inversion. (d) Visualization of the difference in Gaussian noise between steps.

accumulated drifts that significantly degrade reconstruction accuracy and editing fidelity. These drifts arise because the noise latent at the current timestep is unavailable and it has to be approximated using the noise latents from the previous timesteps. The approximated latent is then used to estimate the velocity field for inversion, resulting in shifted velocity fields. As the process continues, the drifts accumulate, leading to significant distortions in the final noise.

Existing RF-based methods [17, 2, 13, 22, 25] aim to reduce the drift to improve editing fidelity. For example, RF-inversion [17] employs two conditional velocity fields: one for inversion (conditioned on Gaussian noise) and one for editing (conditioned on the source image). However, its global application of guidance ignores region specificity, often degrading edit quality. Subsequent methods [2, 22, 13], such as RF-solver [22], refine the discretization process using higher-order ordinary differential equation (ODE) solvers, often combined with attention injection during editing to enhance fidelity. More recently, FTEdit [25] reduces errors by performing iterative average at each inversion timestep, increasing the number of sampling steps. While these approaches reduce errors through finer discretization of the reverse ODE process, they incur additional Neural Function Evaluations (NFEs), decreasing the efficiency. Despite offering slight improvements over RF inversion, these methods remain inefficient and struggle to mitigate the accumulation and amplification of drift.

Actually, existing editing methods overlook the unique properties of RF. Unlike DM, RF models the generation process as a straight trajectory between noise and image, allowing noise latent to be derived via linear interpolation at each timestep. Keeping this in mind, let us revisit the editing process. Note that inversion aims to estimate a noise sample that corresponds to the reference image, enabling an accurate reconstruction of it during sampling. In RF, a better noise sample means a more direct path between the noise and the reference image. Based on this, an intuitive question is: *can we directly refine the desired noise sample in the Gaussian noise domain, rather than gradually transforming the image into noise?* To answer this question, we propose a novel method in this work, called **D**irect **N**oise **A**lignment **Edit**ing (**DNAEdit**). Unlike gradually transforming an image back to Gaussian noise, *DNAEdit directly refines the randomly initialized noise in the Gaussian noise domain, gradually aligning it with a better target noise sample*, as shown in Fig. 1 (b).

To achieve Direct Noise Alignment (DNA), we start from a Gaussian noise sample and iteratively interpolate between it and the clean reference image at each timestep. As shown in Fig. 1 (b), the predicted velocity field often deviates from the expected one derived from linear interpolation. DNA, thus, corrects it by feeding the deviation back into the noise, refining the trajectory. We visualize the changes during the DNA process in Fig. 1 (d). These changes contain the structure of the original image, indicating that the original content is gradually infused into the initial random noise, resulting structured noise. This process produces straighter paths and reduces the accumulated error by avoiding dependence on previous random noise and latents. Finally, the noise converges to a sample that is well aligned with the reference image, as verified by the reconstruction MSE in

Fig. 1(c). We also provide a theoretical analysis showing that DNA shares the same principle with RF-based methods in the **Appendix**.

To better balance editability and fidelity, we introduce a Mobile Velocity Guidance (MVG) to guide denoising by computing the difference between source- and target-conditioned velocity fields in the image domain, which yields a smooth transition from source to target images. In addition, existing benchmarks [6, 10, 28, 16] focus on short prompts, limiting semantic richness. To address this, we introduce **DNA-Bench**, a long-prompt benchmark to evaluate RF editing under detailed textual guidance. Experiments on PIE-Bench and DNA-Bench show that our DNAEdit method strikes a better balance between fidelity and editability, demonstrating the best performance.

## 2 Related Work

**Inversion for Image Editing.** Inversion seeks to transform an image into a corresponding Gaussian noise. A successful inversion generates an initial noise vector that can accurately recreate the reference image, which is essential for further editing. According to DDIM [18], the initial noise can be gradually obtained by iteratively adding predicted noise, but the approximation error can accumulate over timesteps. Various strategies have been developed to address this issue. Null-text inversion [15] optimizes null text embeddings at each inversion step, but it is inefficient. Negative-prompt-inversion improves efficiency by replacing null texts with negative prompts. Unlike DMs, RF has a notably straighter generation path. However, current RF inversion methods still adhere to DM principles, reversing the ODE to gradually add noise. RF inversion [17] aims to tackle this issue by optimizing the reverse ODE process using dynamic optimal control through a linear quadratic regulator to balance editability and fidelity. RF-solver [22] and FireFlow [2] use higher-order solvers to better approximate the reverse ODE and reduce errors at each step. FEEdit [7] employs a fixed-point iteration strategy, refining the added noise and averaging it to suppress approximation errors. Although these approaches can reduce reconstruction error to some extent, they all follow the iterative re-noising paradigm, making error accumulation inevitable. In contrast, our proposed DNA framework directly aligns the noisy latent with the ideal noise derived from model priors within the noise space, significantly reducing the error accumulation.

**Rectified Flow-based Editing.** Compared to DM-based editing methods [1, 21, 4, 5], RF-based editing methods are not fully explored. Unlike most DM models that use the U-Net architecture, RF models [8, 3] primarily use the MM-DiT architecture, which makes strategies relying on U-Net's cross-attention map unsuitable for RF-based editing methods. Some approaches [2, 22] adapt the attention injection scheme to maintain fidelity during editing. Other methods [25] focus on the exchange of text-image information in MM-DiT, manipulating features through AdaLN to control the editing process. While these methods have achieved certain success, they often require additional adaptations for different RF models, limiting their applicability. Some model-agnostic methods have also been developed. FlowEdit [7] uses an inversion-free approach by calculating the difference between the source and the target velocity fields, enabling direct image editing in the image space. However, this method relies on editing the velocity difference of the reference image and restricts the range of editable space, making it difficult to perform global editing tasks (*e.g.*, changing style). Our DNAEdit employs two model-agnostic processes: DNA and MVG. DNA minimizes cumulative error and obtain a better initial Gaussian noise aligned with the source text. MVG guides the editing process to preserve background fidelity while minimizing compromise on editability.

## 3 Direct Noise Alignment Editing (DNAEdit)

### 3.1 Approximation Errors in RF Inversion

**Preliminary.** Rectified Flow (RF) [12, 11] models the transition between two observed distributions $\pi_0$ and $\pi_1$ using an ordinary differential model (ODE):

$$\mathrm{d}Z_t = v_\theta(Z_t)\mathrm{d}t, \quad Z_0 \sim \pi_0, \ Z_1 \sim \pi_1, \ t \in [0, 1], \tag{1}$$

where $v_\theta(\cdot)$ is the learnable velocity field parameterized by $\theta$. To encourage a near-linear trajectory of the transition, RF employs the following objective to train $v_\theta$:

$$\min_\theta \int_0^1 \mathbb{E}[\|(Z_1 - Z_0) - v_\theta(Z_t)\|^2]\mathrm{d}t, \quad Z_t = tZ_1 + (1 - t)Z_0, \tag{2}$$

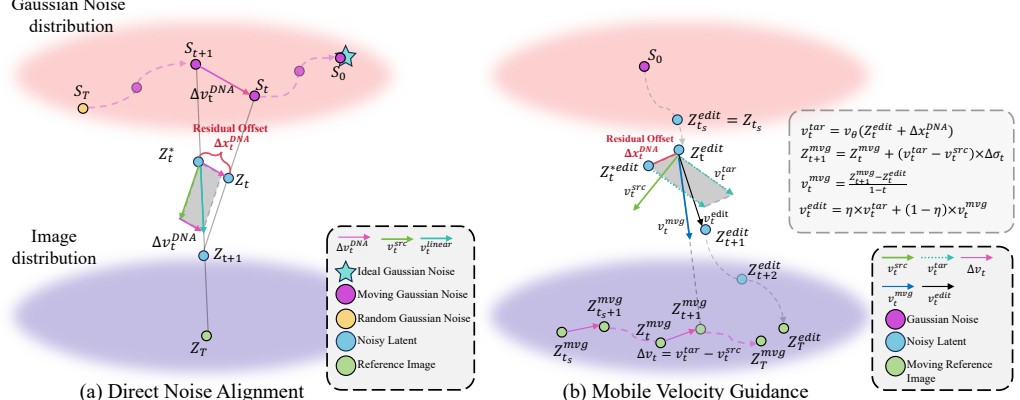

Figure 2: Illustration of (a) DNA and (b) MVG, which collectively build our DNAEdit algorithm.

where $Z_t$ denotes the linear interpolation between $Z_0 \sim \pi_0$ and $Z_1 \sim \pi_1$.

**Approximation Errors in Inversion.** The denoising process maps the standard Gaussian distribution ($\pi_0 = \mathcal{N}(0, 1)$) to the image distribution $\pi_1$, while its reverse maps $\pi_1$ back to $\pi_0$. To solve the ODE in Eq. (1), the time interval $[0, 1]$ is discretized into $T$ steps, denoted as $\{\sigma_0, \ldots, \sigma_T\}$. The Euler solver is used to approximate the solution, with the forward and reverse steps given by:

$$\textbf{Fwd}: Z_{t+1} = Z_t + v_\theta(Z_t)(\sigma_{t+1} - \sigma_t), \quad \textbf{Inv}: Z_t = Z_{t+1} - v_\theta(Z_t)(\sigma_{t+1} - \sigma_t), \quad (3)$$

where the timestep index increases from $0$ to $T$ in the forward process and decreases from $T$ to $0$ in the reverse process. It is evident that in the reverse process, at each step only the noisy latent $Z_{t+1}$ is available, while $Z_t$, the desired output, is unknown. Consequently, the velocity field $v_\theta(Z_t)$ in Eq. (3) cannot be directly evaluated, making the exact inversion intractable.

Considering that the differences between the noisy latents of adjacent timesteps are relatively small, existing RF inversion methods [2, 22, 17, 25] approximate the velocity field at timestep $t$ by evaluating it at timestep $t + 1$, leading to the following inversion formula:

$$\textbf{Inv approx}: v_\theta(Z_t) \approx v_\theta(Z_{t+1}), \quad Z_t \approx Z_{t+1} - v_\theta(Z_{t+1})(\sigma_{t+1} - \sigma_t). \quad (4)$$

However, due to the sequential nature of the inversion process, approximation errors at each step will accumulate over time, resulting in a final latent that can deviate significantly from the expected Gaussian noise (see Fig. 1 (a)). Furthermore, the noise sample obtained through this approximated inversion is not guaranteed to follow the standard Gaussian distribution, mismatching with the model assumption. As a result, the reconstructed or edited image (see Fig. 1 (c)) may be distorted or invalid.

### 3.2 Direct Noise Alignment (DNA)

Our DNAEdit method consists of two key components: DNA and MVG (see Section 3.3). DNA is used to obtain a structured noise sample, enhancing the fidelity of the editing process. The resulting noise is then re-denoised under the guidance of MVG to generate the final image. From the analysis in Section 3.1, we see that the primary source of error comes from the use of the noisy latent $Z_{t+1}$ to replace $Z_t$ to compute the velocity field: $v_\theta(Z_t) \approx v_\theta(Z_{t+1})$. Since the noise levels of $Z_t$ and $Z_{t+1}$ are different, this approximation introduces inconsistencies with the model prior.

Using RF, fortunately, the latent variables $Z_t$ can be constructed via linear interpolation between the clean image and Gaussian noise. One can sample a noise $S \sim \mathcal{N}(0, 1)$ and interpolate it with the image. However, since $S$ is randomly sampled, the resulting path may lead to reconstruction errors under text guidance. According to Eq. (2), there exists an optimal noise such that the RF-driven trajectory forms a nearly linear path to the reference image. Motivated by this, we propose *directly shifting the random noise toward the target noise in the Gaussian space over timesteps:* $S_T \sim \mathcal{N}(0, 1) \rightarrow S_{T-1} \rightarrow \cdots \rightarrow S_0$.

To achieve this goal, we need to devise an effective algorithm to optimize random Gaussian noise $S_T$. This can be done by aligning the velocity $v_t^{src}$ predicted by the RF model with the expected

**Algorithm 1** Direct Noise Alignment (DNA)

---

**Input**: Number of optimization steps $T$, Source image $Z_T$, Source text $\psi^{src}$, Timesteps $\{\sigma_t\}_{t=T}^0$, RF model $v_\theta$ with parameters $\theta$, Randomly sampled start noise $S_T \sim \mathcal{N}(0, 1)$.

**Output**: Noisy latents $\{Z_t\}_{t=T-1}^0$ and residual offset $\{\Delta x_t^{\text{DNA}}\}_{t=T-1}^0$, Terminal noise sample $S_0$

**for** $t = T - 1, ..., 1, 0$ **do**

    $Z_t^* \leftarrow Z_{t+1} \times \frac{\sigma_t}{\sigma_{t+1}} + S_{t+1} \times (1 - \frac{\sigma_t}{\sigma_{t+1}})$              ▷ Interpolate to get noisy latent $Z_t^*$

    $v_t^{\text{linear}} = (S_{t+1} - Z_{t+1})/\sigma_{t+1}$          ▷ Calculate the expected velocity on the linear path

    $v_t^{src} = v_\theta(Z_t^*, \psi^{src})$          ▷ Calculate the predicted velocity by SD3 or FLUX

    $\Delta v_t^{\text{DNA}} = v_t^{\text{linear}} - v_t^{src}$

    $S_t \leftarrow S_{t+1} + \Delta v_t^{\text{DNA}} \times \sigma_{t+1}$          ▷ Move Gaussian noise from $S_{t+1}$ to $S_t$

    $Z_t \leftarrow Z_t^* + \Delta v_t^{\text{DNA}} \times (\sigma_{t+1} - \sigma_t)$          ▷ Move interpolated latent from $Z_t^*$ to $Z_t$

    $\Delta x_t^{\text{DNA}} = Z_t^* - Z_t$          ▷ Compute the residual offset

**end for**

**return** $S_0$, $\{Z_t\}_{t=T-1}^0$ and $\{\Delta x_t^{\text{DNA}}\}_{t=T-1}^0$

---

velocity $v_t^{\text{linear}}$ along the linear path from Gaussian noise to image latent. Specifically, at timestep $t$, we construct the linear path from noise $S_{t+1}$ to latent $Z_{t+1}$ and obtain latent $Z_t^*$ by interpolation:

$$Z_t^* = \frac{\sigma_t}{\sigma_{t+1}} \times Z_{t+1} + (1 - \frac{\sigma_t}{\sigma_{t+1}}) \times S_{t+1}. \tag{5}$$

The estimated noisy latent $Z_t^*$ is obtained via direct interpolation, eliminating the need to approximate it using Eq. (4). We then compute the velocity field $Z_t^*$ using the RF model as $v_t^{\text{src}} = v_\theta(Z_t^*, \psi^{\text{src}})$, and define the expected velocity along the linear path as $v_t^{\text{linear}} = (S_{t+1} - Z_{t+1})/\sigma_{t+1}$. However, as shown in Fig. 2(a), a discrepancy arises between $v_t^{\text{linear}}$ and $v_t^{\text{src}}$ due to the offset between the actual noise $S_{t+1}$ and the noise $S_t$ derived by the predicted velocity. We denote this mismatch as the velocity gap $\Delta v_t^{\text{DNA}}$, and correct it by shifting the noise from $S_{t+1}$ to $S_t$:

$$\Delta v_t^{\text{DNA}} = v_t^{\text{linear}} - v_t^{src}, \quad S_t = S_{t+1} + \Delta v^{\text{DNA}} \times \sigma_{t+1}. \tag{6}$$

As the noise moves to $S_t$, the interpolated noisy latent $Z_t^*$ in Eq. (5) can be refined to a better estimate $Z_t = \frac{\sigma_t}{\sigma_{t+1}} \times Z_{t+1} + (1 - \frac{\sigma_t}{\sigma_{t+1}}) \times S_t$. It can be easily derived that the updated linear velocity $v_t^{\text{linear}} = (S_t - Z_{t+1})/\sigma_{t+1}$ matches the predicted velocity $v_\theta(Z_t^*, \psi^{src})$. The difference between the updated latent $Z_t$ and the initial estimate $Z_t^*$ can be expressed as:

$$Z_t - Z_t^* = \frac{\sigma_{t+1} - \sigma_t}{\sigma_{t+1}} \times (S_t - S_{t+1}). \tag{7}$$

By substituting Eq. (6) into Eq. (7), we can derive that $Z_t = Z_t^* + \Delta v^{\text{DNA}} \times (\sigma_{t+1} - \sigma_t)$. Eq. (7) implies that although the random Gaussian noise $S_T$ may initially deviate from the ideal Gaussian noise, the difference will become much smaller for interpolated latent $Z_t$ due to the small scaling coefficient $(\sigma_{t+1} - \sigma_t)/\sigma_{t+1}$, where $\sigma_{t+1} : 1 \to 0$. Therefore, it can be deduced that $v_\theta(Z_t, \psi^{src}) \approx v_\theta(Z_t^*, \psi^{src})$. Starting from $Z_t$, the predicted velocity field can lead to $Z_{t+1}$ with a small error. Eq. (7) suggests aligning from large to small timesteps to avoid early errors, so we iteratively adjust noise from $\sigma_T$ to $\sigma_0$. We also show that the residual offset $\Delta x_t^{\text{DNA}} = Z_t^* - Z_t$ allows exact reconstruction. Adding it back to $Z_t$, we recover $Z_t^*$ and compute $v_\theta(Z_t^*, \psi^{\text{src}})$ for precise updates from $Z_t$ to $Z_{t+1}$. The algorithm is presented in Algorithm 1, and a theoretical analysis between DNA and existing RF-based methods is provided in the **Appendix**.

### 3.3 Mobile Velocity Guidance (MVG)

Although DNA can estimate a noise sample with small reconstruction error, directly using target texts to guide image generation may destroy the original structure of the reference image. Inspired by [17], we can control the fidelity of the edited image by integrating the velocity field pointed to the reference image. However, if the velocity field is introduced improperly, it can interfere with the denoising process, resulting in *an undesired overlay of the reference and target images* and degrading editability. To address this, we introduce *Mobile Velocity Guidance* (MVG), as illustrated in Fig. 2(b), which adaptively guides the editing process to balance fidelity and editability.

The denoising process starts from a noise sample and gradually moves it to the target image, forming a trajectory from Gaussian noise to the target, called $Z_t^{\text{edit}}$. Specifically, at timestep $t$, we first

---

**Algorithm 2** Mobile Velocity Guidance (MVG)

---

**Input**: Source image $Z_T$, Timesteps $\{\sigma_t\}_{t=0}^{T}$, RF $v_\theta(\cdot)$, Noisy latents $\{Z_t\}_{t=T-1}^{0}$, Residual gaps $\{\Delta x_t^{\mathrm{DNA}}\}_{t=T-1}^{0}$, Start Step $t_s$, Target text $\psi^{tgt}$

**Output**: Edited Image $Z_T^{\mathrm{edit}}$

**Init**: $Z_{t_s}^{edit} = Z_{t_s}$, $\quad Z_{t_s}^{\mathrm{mvg}} = Z_T$

**for** $t = t_s, t_s + 1, \cdots, T - 1$ **do**

$\quad v_t^{tgt} = v_\theta(Z_t^{*\mathrm{edit}}, \psi^{tgt}), \quad Z_t^{*\mathrm{edit}} \leftarrow Z_t^{\mathrm{edit}} + \Delta x_t^{\mathrm{DNA}}$ $\qquad$ ▷ Involve residual gap to get target velocity

$\quad v_t^{src} = v_\theta(Z_t^*, \psi^{src}) = \frac{Z_{t+1} - Z_t}{\sigma_{t+1} - \sigma_t}, \quad \Delta v_t = v_t^{tgt} - v_t^{src}$ $\qquad$ ▷ Reuse source velocity from DNA latents

$\quad Z_{t+1}^{\mathrm{mvg}} \leftarrow Z_t^{\mathrm{mvg}} + \Delta v_t \times (\sigma_{t+1} - \sigma_t)$ $\qquad$ ▷ Move the mobile latent toward the target image

$\quad v_t^{\mathrm{mvg}} = (Z_t^{\mathrm{edit}} - Z_{t+1}^{\mathrm{mvg}})/(1 - \sigma_t)$ $\qquad$ ▷ Calculate the mobile velocity guidance

$\quad v_t^{\mathrm{edit}} = \eta \times v_t^{tgt} + (1 - \eta) \times v_t^{\mathrm{mvg}}$ $\qquad$ ▷ Calculate the synthesized denosing velocity

$\quad Z_{t+1}^{\mathrm{edit}} \leftarrow Z_t^{\mathrm{edit}} + v_t^{\mathrm{edit}} \times (\sigma_{t+1} - \sigma_t)$ $\qquad$ ▷ Perform denoising step

**end for**

**return** $Z_T^{\mathrm{edit}}$

---

incorporate $\Delta x_t^{\mathrm{DNA}}$ to shift $Z_t^{\mathrm{edit}}$ to $Z_t^{*\mathrm{edit}}$, enabling the exact computation of target velocity $v_t^{tgt}$:

$$Z_t^{*\mathrm{edit}} = Z_t^{\mathrm{edit}} + \Delta x_t^{\mathrm{DNA}}, \quad v_t^{tgt} = v_\theta(Z_t^{*\mathrm{edit}}, \psi^{tgt}). \tag{8}$$

This step is crucial. According to Eq. (7), the residual $\Delta x_t^{\mathrm{DNA}} = Z_t^* - Z_t$ is equivalent to the weighted difference between two Gaussian noise samples and therefore free from image content. Essentially, this operation ensures that the regions intended to be preserved in $Z_t^{*\mathrm{edit}}$ are perfectly aligned with the noisy latent in the reverse process, *i.e.*, $Z_t^*$ in Eq. (5). As a result, the computed target and source velocities ($v_t^{tgt}$ and $v_t^{src}$) become near-identical in those preserved regions, which not only improves reconstruction accuracy but also significantly enhances editing fidelity. Our ablation study in **Appendix** further supports this finding.

In addition, there exists another trajectory that transitions from the source image to the target image purely within the image space, denoted as $Z_t^{\mathrm{mvg}}$. Before using $v_t^{\mathrm{tgt}}$ to update the noisy latent $Z_t^{\mathrm{edit}}$ conditioned on the target text, we first apply the velocity difference between the source and target to shift $Z_t^{\mathrm{mvg}}$ from the reference image toward the target image. This velocity difference and the corresponding mobile latent modification are defined as:

$$\Delta v_t = v_t^{tgt} - v_t^{src}, \quad Z_{t+1}^{\mathrm{mvg}} = Z_t^{\mathrm{mvg}} + \Delta v_t \times (\sigma_{t+1} - \sigma_t). \tag{9}$$

Here, we use the saved latents in DNA to calculate the velocity $v_t^{src}$, which not only reduces the number of function evaluations (NFEs) but also accurately captures the velocity field associated with the reconstruction of the reference image. The velocity difference $\Delta v_t$ is then applied to the mobile reference image $Z_t^{\mathrm{mvg}}$, modifying specific regions to reflect the differences induced at timestep $t$ under the source and target text conditions. Using the updated latent $Z_{t+1}^{\mathrm{mvg}}$, we obtain the final denoising velocity by blending the velocity fully conditioned on the target text $v_t^{\mathrm{tgt}}$ with the MVG $v_t^{\mathrm{mvg}}$ using a weighting coefficient $\eta$ as :

$$v_t^{\mathrm{mvg}} = (Z_t^{\mathrm{edit}} - Z_{t+1}^{\mathrm{mvg}})/(1 - \sigma_t), \quad v_t^{\mathrm{edit}} = \eta \times v_t^{tgt} + (1 - \eta) \times v_t^{\mathrm{mvg}}, \tag{10}$$

where $\eta \in [0, 1]$ controls the trade-off between fidelity and editability.

Finally, we apply the synthesized velocity to perform the denoising by $Z_{t+1}^{\mathrm{edit}} = Z_t^{\mathrm{edit}} + v_t^{\mathrm{edit}}(\sigma_{t+1} - \sigma_t)$. This process is applied across all timesteps, as shown in Fig. 2(b). To avoid excessive changes during editing, we skip the initial step and select $Z_{t_s}$ as the starting point for editing. The complete algorithm of MVG is summarized in Algorithm 2. Using a fixed reference image for guidance, as in [17], we maintain the denoising direction close to the reconstruction of the source image during the early editing stages, thereby enhancing the fidelity. However, in later stages, as the image content undergoes substantial changes, relying on the original reference image can severely hinder editability. In contrast, our proposed MVG $v_t^{\mathrm{mvg}}$ mitigates this issue by employing a moving reference for guidance, effectively balancing fidelity and editability throughout the editing process.

Table 2: Quantitative comparison on PIE-Bench. Red, blue, yellow represents top 3 performers.

| Method | Model | Structure | Background Preservation | | | | CLIP Similarity | | Rank |
|---|---|---|---|---|---|---|---|---|---|
| | | Distance ↓ | PSNR ↑ | LPIPS ↓ | MSE ↓ | SSIM ↑ | Whole ↑ | Edited ↑ | Avg. ↓ |
| PnP[21] | SD1.5 | 27.35 | 22.31 | 112.76 | 82.95 | 79.25 | 25.41 | 22.52 | 9.17 |
| MasaCtrl[1] | SD1.4 | 27.12 | 22.19 | 105.44 | 86.37 | 79.91 | 24.03 | 21.15 | 10.42 |
| DI+PnP[6] | SD1.5 | 23.35 | 22.46 | 105.51 | 79.94 | 79.88 | 25.49 | 22.64 | 6.83 |
| DI+MasaCtrl[6] | SD1.4 | 23.58 | 22.68 | 87.41 | 80.63 | 81.51 | 24.39 | 21.41 | 8.75 |
| InfEdit[26] | LCM | 19.31 | 27.31 | 56.32 | 47.80 | 85.30 | 24.90 | 22.14 | 5.67 |
| InsP2P[4] | InsP2P | 58.13 | 20.80 | 159.23 | 221.3 | 76.47 | 23.61 | 21.68 | 13.42 |
| RF-Inv[17] | FLUX | 42.29 | 20.20 | 179.54 | 139.2 | 69.91 | 24.57 | 22.20 | 12.42 |
| RFEdit[22] | FLUX | 21.79 | 24.83 | 113.15 | 52.46 | 83.38 | 25.57 | 22.26 | 6.42 |
| FireFlow[2] | FLUX | 29.03 | 23.33 | 133.40 | 70.83 | 81.22 | 26.19 | 22.99 | 7.42 |
| FlowEdit[7] | FLUX | 27.82 | 21.96 | 112.19 | 94.99 | 83.08 | 25.25 | 22.58 | 9.50 |
| FlowEdit[7] | SD3 | 27.12 | 22.22 | 104.12 | 85.96 | 93.22 | 26.53 | 23.57 | 5.58 |
| FTEdit[25]* | SD3.5 | 18.17 | 26.62 | 80.55 | 40.24 | 91.50 | 25.74 | 22.27 | 3.50 |
| DNAEdit (Ours) | FLUX | 18.87 | 24.99 | 95.06 | 50.45 | 85.71 | 25.79 | 22.87 | 3.42 |
| DNAEdit (Ours) | SD3.5 | 14.19 | 26.66 | 74.57 | 32.76 | 88.63 | 25.63 | 22.71 | 2.50 |

# 4 Experiments

## 4.1 Experimental Settings

**DNA-Bench**. Text-guided editing has been extensively studied, and evaluation benchmarks [6, 10, 16] have been proposed to evaluate and compare the different editing methods. However, most of these benchmarks are developed in conjunction with diffusion-based models. Due to the limitations of text encoders and pre-trained models, diffusion models struggle to accurately understand long text inputs. As a result, existing benchmarks typically feature short descriptions. For example, PIE-Bench [6] has an average prompt length of **9.46** words. In contrast, RF-based models have shown significant improvements in understanding long text input, but the short descriptions in existing benchmarks cannot fully evaluate the editing capabilities of RF-based models.

To bridge this gap, we propose an extended version of PIE-Bench, called DNA-bench, which is tailored for long-text prompts. To construct DNA-Bench, we design target-aware prompts and leverage the powerful multimodal large language model GPT-4o [20] to generate detailed descriptions of the source images as source prompts. In addition, we modify and extend the original target prompts to align with the editing tasks. The average prompt length in DNA-bench is **33.17** words. More details of the construction process and example prompts of DNA-Bench can be found in the **Appendix**.

**Implementation and Compared Methods.** Two versions of DNAEdit are provided, which are based on FLUX-dev [9] and SD3.5-medium [19], respectively. In both versions, the MVG coefficient $\eta$ is fixed at 0.8. Detailed hyper-parameter settings can be found in the **Appendix**. We compare DNAEdit with representative DM-based methods [1, 21, 26] and latest RF-based editing methods [17, 22, 2, 7, 25] using their official implementations and default settings in a shared environment, except FTEdit, where we use provided results.

**Evaluation and Metrics.** We first conduct **reconstruction** and **text-guided editing** experiments on the PIE-bench [6]. The PIE-bench consists of 700 natural and artificial images to evaluate editing methods across 9 distinct dimensions. It provides the source and target prompts for each image, along with the editing area masks to assess background preservation and local editing ability. We then conduct experiments on our proposed DNA-bench. To evaluate the reconstruction and preservation performance of non-edited areas, we adopt the commonly used image quality metrics, including

Table 1: Quantitative results on reconstruction.

| Method | NFE↓ | MSE↓ | LPIPS↓ | SSIM↑ |
|---|---|---|---|---|
| Vallina Inversion | 60 | 0.028 | 0.342 | 0.601 |
| RF-Inversion [17] | 56 | 0.023 | 0.279 | 0.526 |
| RFEdit [22] | 60 | 0.022 | 0.244 | 0.677 |
| FireFlow [2] | 57 | 0.015 | 0.200 | 0.726 |
| DNA | 56 | **0.010** | **0.110** | **0.830** |

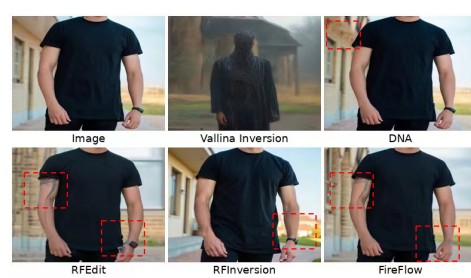

Figure 3: Qualitative comparison on reconstruction.

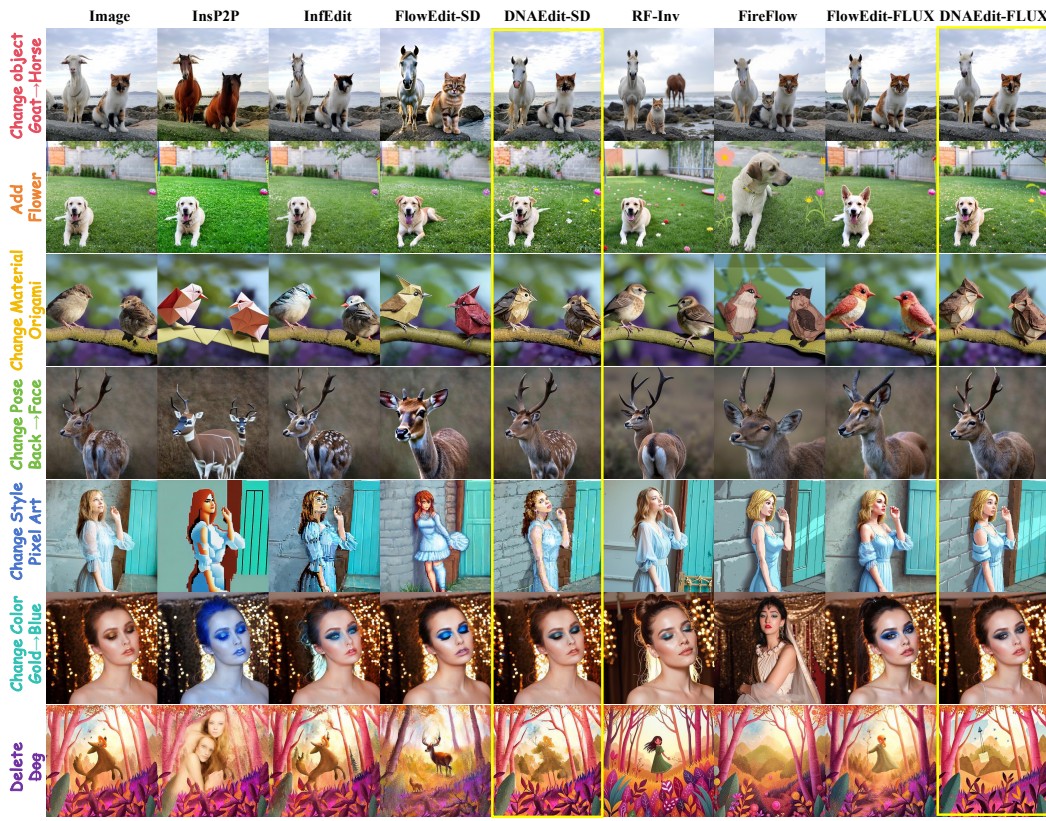

Figure 4: Qualitative comparison of different text-guided image editing methods.

LPIPS [29], SSIM [23], MSE, PSNR and structure distance [6]. Meanwhile, CLIP similarity [24] is employed to assess the consistency between target prompts and edited results.

## 4.2 Main Results

**Results on Reconstruction.** In Fig. 3, we present a qualitative comparison among different RF-based reconstruction methods. We see that our DNA method surpasses other approaches in reconstruction accuracy, with only minor differences from the original image in the area highlighted by red-box. FireFlow and RFEdit both use higher-order ODEs to reduce inversion drift, achieving good overall accuracy. However, noticeable differences appear in areas such as the background and arms. RF-Inversion uses the original image as a reference during reconstruction, leading to over-generation, such as the house in the background, which is not faithful to the original image. Moreover, using the original image as a reference can compromise the editability during editing. The quantitative results are reported in Section 4. Compared to existing inversion-based methods [2, 17, 22], our DNA achieves the lowest reconstruction error under similar NFEs.

**Quantitative Comparison on Text-guided Editing.** In Table 2, we evaluate editing methods in three dimensions: structure preservation, background preservation, and clip similarity. We present the overall ranking for each method by averaging the rankings in these dimensions in the last column of Table 2. Our approach demonstrates superior overall performance, particularly in terms of structure and background preservation. Specifically, while InfEdit shows better PSNR and LPIPS metrics, its CLIP similarity is significantly lower than other methods. This suggests that its strong background preservation hinders its editing capabilities. FlowEdit-SD3 achieves a higher CLIP score but shows weaker background and structure preservation compared to our method (PSNR 22.22dB *vs.* 26.66dB), indicating potential over-editing. This is attributed to its use of a large CFG in the generation, which also leads to poorer visual quality, as illustrated in Fig. 4. Lastly, the recently developed FTEdit shows relatively balanced performance. However, our method excels in background and structure preservation, with similar whole CLIP scores (25.63 *vs.* 25.74). In edited regions, our DNAEdit achieves a notably higher CLIP score, 22.71 *vs.* 22.27, indicating that our approach better preserves

Table 3: Quantitative Results on DNA-Bench. Red , blue , yellow represents top 3 performers.

| Method | Model | Structure | Background Preservation | | | | CLIP Similarity | | Rank |
|---|---|---|---|---|---|---|---|---|---|
| | | Distance ↓ | PSNR ↑ | LPIPS ↓ | MSE ↓ | SSIM ↑ | Whole ↑ | Edited ↑ | Avg. ↓ |
| RFEdit | FLUX | 20.18 | 25.13 | 104.92 | 49.46 | 84.01 | 28.34 | 22.99 | 2.92 |
| RF-Inversion | FLUX | 41.92 | 20.18 | 176.40 | 139.69 | 69.96 | 27.98 | 22.89 | 6.83 |
| FlowEdit | SD3 | 30.93 | 21.35 | 118.48 | 103.52 | 81.45 | 28.89 | 23.58 | 4.58 |
| FlowEdit | FLUX | 29.06 | 21.57 | 116.97 | 102.50 | 82.54 | 28.15 | 22.86 | 5.33 |
| FireFLow | FLUX | 26.97 | 23.57 | 124.78 | 68.51 | 81.82 | 28.69 | 23.32 | 3.83 |
| DNAEdit | FLUX | 18.61 | 24.89 | 93.76 | 50.80 | 85.80 | 28.36 | 22.99 | 2.41 |
| DNAEdit | SD3.5 | 25.67 | 23.24 | 112.60 | 67.32 | 83.69 | 28.90 | 23.66 | 2.08 |

non-edited areas while accurately editing targeted regions. Furthermore, our method is model-agnostic and does not require alterations to the MM-DiT architecture, enhancing its applicability.

**Qualitative Comparison on Text-guided Editing.** We present visual comparisons of state-of-the-art editing methods in Fig. 4, including object, attribute, color, material, style, and pose editing. It can be found that InfEdit often results in edits without visible changes (see the 2nd, 4th and 7th rows). In contrast, FireFlow frequently exhibits over-editing, as shown in the 2nd, 4th and 6th rows. Although the editing instructions are followed, the original image's structure is significantly changed. As for FlowEdit, while the SD3 version of it successfully accomplishes most editing tasks without altering much the original image's structure, it suffers from low visual quality and noticeable over-saturation, as evident in row 1 and row 4. The FLUX version of FlowEdit offers better visual quality, but it has some shortcomings in fidelity and instruction following for some editing tasks. For instance, in row 2, it fails to add a flower, and there is a noticeable change in the dog's ears. Compared to these methods, our proposed DNAEdit method is versatile across various editing tasks and shows superior visual quality. For example, in row 5, our approach successfully applies a global pixel style while preserving the original image's character pose and overall structure. Similarly, in row 6, our method effectively changes the makeup color while retaining facial ID and other details, achieving desired results for both edited and non-edited areas. In summary, DNAEdit effectively balances between editability and fidelity, which is consistent with our quantitative results.

**Results on DNA-Bench.** Table 3 shows the results of RF-based editing methods on DNA-Bench. Our DNAEdit remains the best, achieving high clip scores while preserving the background. This validates that DNAEdit can be used for image editing with short- and long-text inputs without any change. Comparing the results of the same method under long and short prompts, we can see that inversion-based methods achieve better background preservation with long prompts (*e.g.*, FireFLow achieves a PSNR of 23.33dB on PIE-Bench and a PSNR of

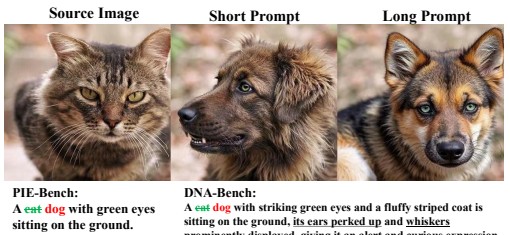

Figure 5: Visual comparison between editing results using short and long prompt.

23.57dB on DNA-Bench). This is because long texts provide detailed descriptions that align with image contents, allowing more accurate reconstruction. As shown in Fig. 5, short prompts change a cat into a dog but alter the pose, while long prompts keep the original pose, producing consistent results. In addition, long prompts also result in improved clip similarity scored.

## 4.3 Ablation on Proposed Modules

**Baseline.** In Table 4, we present the experimental results of our proposed method with different modules. We begin with Exp. ①, which utilizes velocity fields computed on interpolated latents to reverse ODE process. In Exp. ①, we initiate the inversion process by randomly sampling Gaussian noise, which is then used throughout the inversion process. At each step, the Gaussian noise is interpolated with the latent at timestep $t$ to compute the velocity field for timestep $t-1$, facilitating the inversion. During the re-denoise phase, we use the velocity field conditioned on target prompt to generate target image. By employing interpolation to construct the latent of timestep $t-1$, we can mitigate errors caused by approximation. As observed, this setting already achieves relatively good results. However, despite avoiding approximation errors at each step, the use of a fixed Gaussian

Table 4: Ablation study on proposed modules.

| Exp. | Component | Structure | Background Preservation | | | | CLIP Similarity | |
|---|---|---|---|---|---|---|---|---|
| | | Distance ↓ | PSNR ↑ | LPIPS ↓ | MSE ↓ | SSIM ↑ | Whole ↑ | Edited ↑ |
| ① | Interpolation | 31.90 | 22.61 | 147.77 | 84.60 | 78.93 | 25.73 | 22.47 |
| ② | +DNA | 32.67 | 22.18 | 146.27 | 89.92 | 79.73 | 26.24 | 23.08 |
| ③ | +ResOffset | 24.93 | 23.75 | 120.56 | 64.81 | 82.24 | 26.02 | 22.79 |
| ④ | +DNA +ResOffset | 33.98 | 21.97 | 149.84 | 95.19 | 79.32 | 26.40 | 23.23 |
| ⑤ | +DNA+MVG | 18.81 | 24.91 | 95.44 | 50.29 | 85.55 | 25.78 | 22.48 |
| ⑥ | +DNA +ResOffset+MVG | 18.87 | 24.99 | 95.06 | 50.45 | 85.71 | 25.79 | 22.87 |

noise throughout the process can introduce significant errors when approaching pure noise, especially when the initially sampled Gaussian noise deviates much from the desired structure noise.

**Effectiveness of DNA.** In Exp. ②, we introduce DNA by adjusting the initially sampled Gaussian noise during the inversion process. By comparing ① and ②, we can observe that as the structural and background preservation metrics remain nearly unchanged, the Clip Whole and Clip Edited similarity significantly increase, improving from 25.73 and 22.47 to 26.24 and 23.08, respectively. This improvement is attributed to the DNA process, where the noise is continuously moved by the difference between the linear velocity and the velocity conditioned on the source prompt. This movement injects the original image structure and gradually aligns the random noise with the structured noise corresponding to source prompt, enabling the generated results to better match the target prompt during editing.

**Effectiveness of ResOffset.** Comparing Exp. ① and ③, we see that the introduction of ResOffset during the re-denoising process significantly improves the metrics for image structure and background preservation. For instance, the Structure Distance improves from 31.90 to 24.93, and the PSNR increases from 22.61 dB to 23.75 dB. By incorporating the latent calculated with ResOffset, we shift $Z_t^{edit}$ to $Z_t^{*edit}$, enhancing the consistency between the intended preserved regions in the noisy latent $Z_t^*$ during the DNA process and those $Z_t^{*edit}$ during re-denoising. This ensures that the velocity field during denoising accordingly tends to preserve these regions.

**Effectiveness of MVG.** Comparing Exp. ④ and ⑥, we observe that the introduction of MVG results in a decrease in structure similarity from 33.98 to 18.87, while PSNR increases from 21.97 dB to 24.99 dB, indicating a significant improvement in editing fidelity. As a trade-off for faithfulness to the original image, there is a decrease in CLIP similarity. Since MVG ensures that the overall structure of the edited image undergoes minimal changes, it imposes greater limitations on whole image editing, leading to a decrease in the whole image metric from 26.40 to 25.79 (-0.61). However, because MVG distinguishes between edited and non-edited regions and uses the evolved image to guide the editing of specific areas, the restriction on edited regions is greatly reduced, resulting in only a slight decrease of 0.36 in Edited CLIP similarity. This demonstrates that under the guidance of MVG, fidelity can be significantly enhanced while avoiding the constraints on edited regions that could lead to unchanged results. Please refer to **Appendix** for detailed ablation studies on the setting of the MVG coefficient.

## 5    Conclusion

We presented a novel RF-based method, namely DNAEdit, for text-guided image editing. By utilizing RF's property of linear trajectory, we proposed a method to estimate accurate latents by calculating the velocity fields at specific timesteps through random sampling and linear interpolation. By analyzing the expected and predicted velocity fields, we presented DNA to align the image to the ideal noise directly in the Gaussian noise domain. We then introduced MVG to maintain background areas while guiding effective changes in editing regions. Theoretical analyses were provided to explain the effectiveness of DNA and MVG and their connections with existing RF-based editing methods. Experiments were conducted on the commonly used PIE-Bench and our newly improved long-text DNA-Bench. Both qualitative and quantitative results showed that our DNAEdit approach performed well on various editing tasks, producing high-quality edits faithful to the original image.

**Limitations**. As a training-free editing method, DNAEdit utilizes the strong priors of pre-trained T2I models by converting images into structured noise aligned with the given text. Therefore, it may fall short in achieving desired editing results for cases that lie outside the foundation T2I model's prior.

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
