# OpenReview forum: "DNAEdit: Direct Noise Alignment for Text-Guided Rectified Flow Editing"
_NeurIPS.cc/2025/Conference — NeurIPS 2025 spotlight_

### Official Review · Reviewer_H5bA · 2025-06-27

**Clarity:** 3
**Significance:** 3
**Originality:** 3
**Rating:** 4
**Confidence:** 4

**Summary:**

The paper focuses on image editing and 3 contrbutions:
1. Instead of running inversion, it directly optimize the latent noise to find a good structed noise that correspondes to the original image. The method is called Direct Noise Alignment (DNA).
2. It proposes a new vector field for editing in the reverse process, called Mobile Velocity Guidance (MVG), which does not rely on a fixed terminal state.
3. A new benchmark DNABench is proposed for image editing with long prompts.

The authors proveded both qualittive and quantitative results to demonstarte the proposed method.

**Questions:**

1. I'm a bit confused with E.3. In my understanding, \eta controls the v_t^{tgt} in eq. 10, while v_t^{MVG} controls the background preservation. This seems to be misalined with the definitions of MVG in E.3?

2. what is the choice of start time t_s?

3. typo in Algo. 1, 3rd to the last row, should be Z_t?

4. Regarding MVG, as moving Z_t^{MVG} to the direction of v_t^{tar} - v_t^{src}, will it give less and less control, eventually converging with v^tgt?

**Ethical Concerns:**

["NO or VERY MINOR ethics concerns only"]

**Final Justification:**

The authors answers addressed most of my concerns. The proposed method maps X_0 directly to noises instead of running inversion. I think this is a novel idea, however the design is a bit over-complicated, the writing can be improved to better illustrate the idea, which is the main reason I keep the score.

**Limitations:**

yes

**Quality:**

3

**Strengths And Weaknesses:**

Strength:

1. The motivation is well illustrated and the methods are clearly written.
2. The authors provide comprehensive expriments, including ablation studies on different component.
3. Both qualitative and quantiative results demonstarte the effectiveness of the proposed method.

Weakness:
1. The constructed vector vield is also a bit over-complicated, I think more detailed ablations on MVG and be futher studied.
2. More explain on why is direct noise alignment a better design choice than inversion?
3. Current image inversion methods have some weaknesses including hyperparaemter robustness and background robuestness. The proposed method mainly demonstrated the second part but the details in hypermarameter choice should be further illustrated.

---

> ### Author Rebuttal · Authors · 2025-07-30
>
> We sincerely thank this reviewer for the constructive comments and suggestion. We hope our following point-to-point responses can address this reviewer's concerns.
>
> **Weakness 1: The constructed vector field is also a bit over-complicated, I think more detailed ablations on MVG can be further studied.**
>
> Thanks for the suggestion. Although the formula looks complex, the underlying idea of our constructed vector field is straightforward. We introduce $ v^{mvg} $, a velocity oriented toward region-specific changes in the original image. This can suppress unintended modifications in background areas.
>
> As suggested, we have conducted additional ablation studies on MVG coefficient $\eta$ below.
> | Exp. | $\eta$ | Structure Distance $\downarrow$ | Background Preservation       |                  |                  |                  | CLIP Similarity       |                  |
> |:----:|:------:|:-------------------------------:|:------------------------------:|:----------------:|:----------------:|:----------------:|:---------------------:|:----------------:|
> |      |        |                                 | PSNR $\uparrow$                | LPIPS $\downarrow$ | MSE $\downarrow$ | SSIM $\uparrow$  | Whole $\uparrow$      | Edited $\uparrow$ |
> |  ①   |  1.0   |             33.98               |             21.97              |      149.84      |      95.19       |      79.32       |       **26.40**       |      **23.23**   |
> |  ②   |  0.9   |             24.52               |             23.64              |      118.20      |      67.05       |      83.06       |        26.16          |       22.98      |
> |  ③   |  0.8   |             18.87               |             24.99              |       95.06      |      50.45       |      85.71       |        25.79          |       22.87      |
> |  ④   |  0.7   |           **15.32**             |           **26.06**            |      **79.43**   |     **39.20**    |     **87.58**    |        25.39          |       22.06      |
>
>
> We can see that
> an excessively large $\eta$ value (e.g., 1), which means denoising fully according to the velocity $v^{tgt}$, will generate new images based on the target prompt. Although the CLIP metrics are high, the fidelity to the original image is significantly reduced. On the other hand, an excessively small $\eta$ value (e.g., 0.7 or lower) will lead to editing failure. Based on our experience, when $\eta$ is set to around 0.8-0.9, it can both maintain the fidelity of the image and ensure the success rate of editing. We will add this study in the revision.
>
>
> **Weakness 2: More explain on why is direct noise alignment a better design choice than inversion?**
>
>
> As shown in Equation (4) in the preliminary section of the main paper, during the conventional noise inversion process, we cannot obtain $ Z_t $ to compute $ v_\theta(Z_t) $ and thus can only use $ Z_{t+1} $ with an incorrect noise level, introducing errors. These errors accumulate gradually during the entire noise inversion process.
>
> In contrast, our DNA method leverages the unique property of Rectified Flow; that is, the latents can be constructed via linear interpolation. We construct $ Z_t $ with the correct noise level through linear interpolation, avoiding errors caused by mismatched noise levels. Moreover, by using velocity differences to directly adjust Gaussian noise within the noise domain, DNA reduces accumulated errors, resulting in a better starting point for editing.
> We will add more discussions in the revision.
>
> **Weakness 3: Current image inversion methods have some weaknesses including hyperparameter robustness and background robustness. The proposed method mainly demonstrated the second part but the details in hypermarameter choice should be further illustrated.**
>
>
> Sorry for missing the details. In our experiments, for the FLUX model, we use 28 optimization steps, set the Classifier-Free Guidance (CFG) scale to 2.5, and set $t_s$ to 4. The coefficient $\eta$ for MVG is set to 0.8. Please refer to Section C.1 in the Appendix for more details of hyperparameter choice.
> In addition, our method allows for fine control over the editing effect of a single image by adjusting only $\eta$ while keeping the CFG and $t_s$ fixed, making the selection of hyperparameters simple.
>
> **Question 1: I'm a bit confused with E.3. In my understanding, $\eta$ controls the $v_t^{tgt}$ in eq. 10, while $v_t^{MVG}$ controls the background preservation. This seems to be misalined with the definitions of MVG in E.3?**
>
> Equation (10) defines $ v_t^{edit} = \eta v_t^{tgt} + (1-\eta) v_t^{mvg} $. In Section E.3, we show that MVG can effectively ensure faithfulness to the reference image. A smaller $\eta$ means that the velocity $ v_t^{edit} $ is closer to $ v_t^{mvg} $, which can better preserve the background of the original image. This is consistent with the formula. Our notation ``MVG=1.0" (instead of $\eta=1.0$) in Figure 12 of the Appendix may cause confusion. This was an oversight in labeling, and we apologize for the misunderstanding caused.  Please feel free to raise further questions during the subsequent discussion session if this explanation does not fully address your concerns.
>
> **Question 2: what is the choice of start time $t_s$?**
>
>
> For the FLUX version, $ t_s $ is set to 4; for the SD version, $ t_s $ is set to 13. For more detailed hyperparameter settings, please refer to Section C.1 in the Appendix.
>
> **Quesstion 3: typo in Algo. 1, 3rd to the last row, should be $Z_t$?**
>
>  We sincerely apologize for this typo. We will carefully double-check all formulas throughout the paper.
>
> **Quesstion 4: Regarding MVG, as moving $Z_t^{MVG}$ to the direction of $v_t^{tar} - v_t^{src}$, will it give less and less control, eventually converging with $v^{tgt}$?**
>
>
> Yes, as $Z_t^{MVG} $ moves, $ Z_t $ also evolves toward the target image, meaning that the velocity $ v^{mvg} $ directing it gradually converges with $ v^{tgt} $.
> We consider this convergence as a reasonably adaptive process: in the initial stage, $ v^{mvg} $ prevents excessive disruption of the image content; in the subsequent stages, it enables content generation aligned with the target prompt to produce the final high-quality edited result.

---

> ### Author Response · Authors · 2025-08-04
>
> Dear Reviewer H5bA,
>
> Many thanks for your time in reviewing our paper and your constructive comments. We have submitted the point-to-point responses. We appreciate if you could let us know whether we have addressed your concerns, and we are happy to answer any further questions.
>
> Best regards,
>
> Authors of paper \#1255

---

> > ### Comment · Reviewer_H5bA · 2025-08-06
> >
> > Thanks the authors for addressing my concerns, the method is novel and the experiments are comprehensive. I think writing can be improved to better illustrate the underlying idea behind the complictaed model designs to help the readers better understand the paper.

---

> > > ### Author Response · Authors · 2025-08-06
> > >
> > > Thank you for your time and valuable feedback. We appreciate your constructive comments and will carefully revise the manuscript to improve clarity and better highlight the key ideas.

---

### Official Review · Reviewer_iQQ3 · 2025-06-29

**Clarity:** 3
**Significance:** 3
**Originality:** 3
**Rating:** 5
**Confidence:** 4

**Summary:**

This paper proposes a text-guided image editing method, DNAEdit, for the rectified flow model. The method is motivated by the limitation of the previous method, which suffers from estimation error during the reverse process. To tackle this problem, this paper proposes the Direct Noise Alignment (DNA) to optimize the random noise in the Gaussian space, and the Mobile Velocity Guidance (MVG) to apply editing while preserving the unchanged regions.  The effectiveness is evaluated on the PIE-Bench and a new proposed DNA-Bench considering longer text inputs.

**Questions:**

Would initializing the Gaussian noise by obtaining from previous methods, following **Eq.4** rather than random noise, lead to better performance when combined with the proposed DNA approach?

**Ethical Concerns:**

["NO or VERY MINOR ethics concerns only"]

**Final Justification:**

I have read the rebuttals, and my concerns are addressed. I keep my positive score and recommend accepting this paper.

**Limitations:**

yes

**Quality:**

3

**Strengths And Weaknesses:**

### Strengths
- The proposed method is well motivated. The estimation error during reverse is a key challenge for image editing.
- The contribution is solid, including both a novel editing method from the perspective of latent noise optimization and an evaluation benchmark.
- The paper is well written and easy to follow.

### Weaknesses
- As the optimization starts from a randomly initialized Gaussian noise $S_{t+1}$, it raises concerns about the initial estimation error of
$Z_{t}^{*}$.  Is there a possibility that suboptimal initializations could hinder convergence to the ideal Gaussian noise?
- The proposed method performs consistently worse than prior methods on the CLIP Similarity metric, as shown in Table 2. Reasons and further analyses are expected for this observation.

---

> ### Author Rebuttal · Authors · 2025-07-30
>
> We sincerely thank this reviewer for the constructive comments and suggestion. We hope our following point-to-point responses can address this reviewer's concerns.
>
> **Weakness 1: As the optimization starts from a randomly initialized Gaussian noise $S_{t+1}$, it raises concerns about the initial estimation error of $Z_t^*$. Is there a possibility that suboptimal initializations could hinder convergence to the ideal Gaussian noise?**
>
> As shown in Equation (7) in the main paper: $Z_t-Z^\star_t=\frac{\sigma_{t+1}-\sigma_{t}}{\sigma_{t+1}}\times (S_{t}-S_{t+1})$, although the random noise $ S_T $ may initially deviate from the ideal Gaussian noise, such deviations caused by random sampling can be significantly reduced in the interpolated latent variable $ Z_t $ due to the small scaling coefficient $ (\sigma_{t+1} - \sigma_t)/\sigma_{t+1} $, where $ \sigma_{t+1} $ ranges from 1 to 0. In our implementation, we did not encounter difficulties in convergence caused by the initial Gaussian noise.
>
> **Weakness 2: The proposed method performs consistently worse than prior methods on the CLIP Similarity metric, as shown in Table 2. Reasons and further analyses are expected for this observation.**
>
>
> In Table 2 of the main paper, when compared with FLUX-based baselines, the  CLIP metrics of our method (CLIP-Whole: 25.79; CLIP-Edited: 22.87) are only slightly lower than those of Fire-Flow (CLIP-Whole: 26.19; CLIP-Edited: 22.99). However, Fire-Flow performs worse than most other methods, including ours, in terms of structural and background preservation metrics. This indicates that Fire-Flow tends to over-edit, *i.e.*,  generating images that align with the target prompt but at the price of preserving the original structure, which artificially inflates its CLIP scores.
> For SD3-based methods, a similar phenomenon can be observed. Although our CLIP metrics are lower than those of FlowEdit-SD, it can be seen that our method's structural distance (14.19) is much lower than that of FlowEdit (27.12), and it has significant gaps in background preservation metrics such as PSNR (26.66 vs 22.22 lower is better). This indicates that FlowEdit also suffers from over-editing. In addition, from the visual comparisons in Fig. 4 of the main paper and Figs. 9 and 10 of the Appendix, it can be seen that the results of FlowEdit-SD have the issue of over-editing. Moreover, FlowEdit requires a relatively large CFG coefficient, which leads to obvious over-saturation and severely reduces visual quality.
>
>
> **Question 1: Would initializing the Gaussian noise by obtaining from previous methods, following Eq.4 rather than random noise, lead to better performance when combined with the proposed DNA approach?**
>
>
> Previous inversion-based methods do not sample noise; instead, they start from a clean image and estimate noise step by step to obtain a Gaussian noise, *i.e.*, the so-called noise inversion. In contrast, our method directly starts from a random Gaussian noise, so it does not need to utilize previous methods for noise initialization. Following this reviewer's suggestion, we conducted an experiment using the inverted noise by FireFlow as the initial noise of our DNA process.
> The results are shown in the table below. We can see that the two initialization methods show almost no difference, but this inversion-based initialization introduces an additional 50\% computational load. This experiment can also demonstrate that our DNA method is insensitive to the randomness introduced by initialization—any sampled Gaussian noise can be optimized to the ideal noise.
>
> | Exp.                     | Structure Distance $\downarrow$ | Background Preservation       |                  |                  |                  | CLIP Similarity       |                  |
> |--------------------------|:-------------------------------:|:------------------------------:|:----------------:|:----------------:|:----------------:|:---------------------:|:----------------:|
> |                          |                                 | PSNR $\uparrow$                | LPIPS $\downarrow$ | MSE $\downarrow$ | SSIM $\uparrow$  | Whole $\uparrow$      | Edited $\uparrow$ |
> | Inversion Noise Initilization |             19.73               |             24.95              |      **94.47**   |      50.47       |     **85.78**    |        25.77          |       22.39      |
> | Random Gaussian Sampling  |           **18.87**             |           **24.99**            |       95.06      |     **50.45**    |      85.71       |       **25.79**       |      **22.87**   |
> | $\Delta$                 |             -0.14               |             +0.04              |       +0.59      |      -0.02       |      -0.07       |        +0.02          |       +0.48      |

---

> > ### Comment · Reviewer_iQQ3 · 2025-08-04
> >
> > Thanks to the authors for their efforts in rebuttal, which addressed my concerns theoretically and experimentally. Good luck.

---

> > > ### Author Response · Authors · 2025-08-04
> > >
> > > We sincerely thank this reviewer for the time and engagement in reviewing paper, as well as the constructive comments, which are valuable for us to further improve the work.
> > >
> > > Authors of paper \#1225

---

### Official Review · Reviewer_rquf · 2025-06-30

**Clarity:** 4
**Significance:** 4
**Originality:** 3
**Rating:** 5
**Confidence:** 4

**Summary:**

Existing image editing methods (diffusion-based and Rectified flow methods) suffer from noise accumulation during the inversion process. The work proposes DNAEdit, consists of Direct Noise Alignment (DNA) and Mobile Velocity Guidance (MVG). DNA refines the Gaussian noise in the latent space directly by aligning predicted and expected velocity fields at each timestep, thereby reducing reconstruction drift. MVG can balance background preservation and target object editability. Additionally, the authors propose DNA-Bench, a new benchmark for evaluating long-prompt editing tasks.

**Questions:**

Aside from the questions proposed in  "weakness" section, I d like the authors clarify the following questions:
1. The concern about the efficiency of the proposed methods, including the inference speed with baseline.
2. The reason why the DNAedit methods can boost the long prompt editing.
3. The authors show DNAedit can be applied into video editing domain as well, which is promising. Thus, i hope the authors can provide more details and techniques about this part.

**Ethical Concerns:**

["NO or VERY MINOR ethics concerns only"]

**Final Justification:**

Most of my concerns have been addressed in the rebuttal stage.

**Limitations:**

The limitations of the work are mainly as follows:
1. some typos and bad formulations should be improved.
2. more experimental results should be revealed.

**Paper Formatting Concerns:**

I dont find any formatting concerns.

**Quality:**

3

**Strengths And Weaknesses:**

### Weaknesses

- The authors need to carefully recheck the correctness of the formulas to avoid typos. For example, in Algorithm 1, the calculation of the residual offset should be $Z_t^*−Z_t$​. These formulas are crucial to the method's implementation.
- I would like to know where the superiority and uniqueness of the design of Equation (5) lie. Have the authors considered and compared other interpolation methods? For example, an alternative formulation could be $Z_t^* = \sigma_t \times Z_T + (1-\sigma_t) \times S_{t+1}$. Compared to such a formulation, only $S_{t+1}$ is inaccurate in this case, whereas in the original Equation (5), both $S_{t+1}$ and $Z_{t+1}$ are approximated. Could the authors elaborate more on the motivation behind their design and its advantages?
- Could the authors please provide more specific details regarding the experimental settings, including the number of optimization steps $T$, the Classifier-Free Guidance scale, and any other relevant hyperparameters? This information would be helpful for better understanding and reproducing the results.

### strength

- This paper proposes a novel DNA inversion method, which gradually aligns randomly sampled Gaussian noise to the ideal inversion noise through an iterative process, achieving excellent reconstruction performance. The method introduces MVG for image editing.
- A long-text prompt benchmark, named DNA-bench, is proposed to evaluate text-driven image editing methods.
- The proposed method achieves comparable performance on PIE-bench and SOTA result on DNA-bench.

---

> ### Author Rebuttal · Authors · 2025-07-30
>
> We sincerely thank this reviewer for the constructive comments and suggestion. We hope our following point-to-point responses can address this reviewer's concerns.
>
> **Weakness 1: The authors need to carefully recheck the correctness of the formulas to avoid typos. For example, in Algorithm 1, the calculation of the residual offset should be $Z_t^\*-Z_t$. These formulas are crucial to the method's implementation.**
>
> Thank you very much for carefully reading our work and pointing out the issue. We are sincerely sorry for this typo. We will carefully check all formulas throughout the paper to ensure their correctness and enhance readability.
>
> **Weakness 2: I would like to know where the superiority and uniqueness of the design of Equation (5) lie. Have the authors considered and compared other interpolation methods? For example, an alternative formulation could be $ Z_t^\* = \sigma_t \* Z_T + (1 - \sigma_t) \* S_{t+1} $. Compared to such a formulation, only $ S_{t+1} $ is inaccurate in this case, whereas in the original Equation (5), both $ S_{t+1} $ and $ Z_{t+1} $ are approximated. Could the authors elaborate more on the motivation behind their design and its advantages?**
>
> We actually considered the interpolation method mentioned by this reviewer, but we finally chose the interpolation scheme presented in the paper. This is because our goal is to align the expected velocity from $ Z_t $ to $ Z_{t+1} $ with the predicted velocity at $ Z_t $. In contrast, the interpolation method mentioned by this reviewer aligns the velocity from $ Z_t $ to $ Z_T $ with the predicted velocity at $ Z_t $. According to the denoising ODE process of the flow matching model, *i.e.*, $ Z_{t+1} = Z_t + v_\theta(Z_t)(\sigma_{t+1} - \sigma_t) $, our presented interpolation scheme reduces the error at each step, minimizing the overall error more effectively during reconstruction.
>
> In addition, our interpolation scheme offers an extra advantage. The residual offset generated during the DNA process enables error-free reconstruction. The details can be found in Sections A.2 and A.3 of the Appendix.
>
> Regarding the question that "in  Equation (5), both $ S_{t+1} $ and $ Z_{t+1} $ are approximated, while in your proposed interpolation scheme, only $ S_{t+1} $ is approximated": by comparing Equation (5) with the interpolation scheme mentioned by this reviewer, we can see that the coefficients for the common term $ S_{t+1} $ are $ 1 - \frac{\sigma_t}{\sigma_{t+1}} $ and $ 1 - \sigma_t $, respectively, which correspond to sequences such as \{1/10, 1/9, 1/8..., 1/2, 1\} and \{1/10, 2/10, 3/10, ..., 9/10, 1\}. Since the initial noise $ S $ is sampled randomly, its error is relatively large. However, the coefficient sequence of our interpolation scheme will rely less on $ S_{t+1} $ when $ S_{t+1} $ is unreliable, therefore it will not increase the error during the noise alignment process.
>
> **Weakness 3: Could the authors please provide more specific details regarding the experimental settings, including the number of optimization steps , the Classifier-Free Guidance scale, and any other relevant hyperparameters? This information would be helpful for better understanding and reproducing the results.**
>
> In our experiments, for the FLUX model, we use 28 optimization steps, set the Classifier-Free Guidance (CFG) scale to 2.5, and set $t_s$ to 4. And for the SD version, we use the SD3.5-medium model, with DNA steps set to 40, $t_s$ set to 13, and CFG set to 3.5. Both versions have the MVG coefficient $\eta$ set to 0.8. Please refer to Section C.1 in the Appendix for more details.
>
> **Question 1: The concern about the efficiency of the proposed methods, including the inference speed with baseline.**
>
> Thanks for the suggestion. We compare the efficiency of our model with the baseline methods in the table below. Compared with DDIM-based methods, RF-based methods have an overall smaller number of sampling steps because of their straighter trajectories. Compared with RF-based methods, our DNAEdit has more NFEs than FireFlow but outperforms other RF-based methods. In terms of inference time, DNAEdit introduces little additional computational cost, as all operations involved are simple calculations. Overall, DNAEdit is comparable to other RF-based methods in terms of efficiency.
>
> | **Methods**       | **Model**       | **NFEs** | **Time(s)** |
> |-------------------|:---------------:|:--------:|:-----------:|
> | DNAEdit-FLUX      | FLUX            | 48       | 8.35        |
> | DNAEdit-SD        | SD3.5-medium    | 54       | 4.05        |
> | RF-Inversion      | FLUX            | 56       | 11.2        |
> | FireFLow          | FLUX            | 32       | 5.9         |
> | RF-Solver         | FLUX            | 60       | 10.84       |
> | FlowEdit-FLUX     | FLUX            | 48       | 8.32        |
> | FlowEdit-SD       | SD3-medium      | 66       | 3.14        |
> | InfEdit           | LCM             | 24       | 1.92        |
> | PnP               | SD 1.5          | 100      | 10.31       |
> | MasaCtrl          | SD 1.4          | 100      | 20.07       |
>
> **Question 2: The reason why the DNAEdit methods can boost the long prompt editing.**
>
> On one hand,  as illustrated in Figure 1 (d) of the main text, the DNA process injects information from the original image into the noise by computing the velocity differences $v_t^{src}-v_t^{linear}$, where $v_t^{src} $ is the velocity conditioned on the source prompt. For long prompts that have more detailed information, the DNA process will inject the original image structure more precisely into the noise, thereby better preserving the information of the original image. On the other hand, in our MVG process, it is important to note that $ v^{mvg} $ is derived from the difference between $ v^{src} $ and $ v^{tgt} $, which are conditioned on the source and the target prompts, respectively. More detailed prompts will make this difference focus more precisely on the editing region. In summary, DNAEdit can better leverage the advantages offered by long prompts, thereby achieving superior editing performance.
>
> **Question 3: The authors show DNAEdit can be applied into video editing domain as well, which is promising. Thus, I hope the authors can provide more details and techniques about this part.**
>
> Our method is model-agnostic. Thus, applying it to video editing can be done by implementing the DNA and MVG processes as we do in image editing.   The key differences lie in that we need to employ a pre-trained text-to-video (T2V) model instead of a text-to-image (T2I) model and adjust the hyperparameters.   In our experiment, we used the Wan 2.1 1.3B T2V model. The number of sampling steps is set to 50 by default. For the DNA process, the CFG scale is 1. For the MVG process, the CFG scale is 5.0 (as recommended by the Wan T2V model), $t_s$ is set to 12, and the MVG coefficient ranges from 0.9 to 1.  Such a simple modification in implementation further demonstrates the generality of DNAEdit for RF-based generative models.

---

> > ### Comment · Reviewer_rquf · 2025-08-05
> > **Response to the rebuttal**
> >
> > Thanks for your rebuttal, most of my concerns have been solved. I believe it is a novel and solid work.

---

> > > ### Author Response · Authors · 2025-08-05
> > >
> > > We are happy to know that the concerns of this reviewer have been addressed. Your time and engagement in reviewing our paper and your constructive comments are greatly appreciated!
> > >
> > > Authors of paper #1225

---

> ### Author Response · Authors · 2025-08-04
>
> Dear Reviewer rquf,
>
> Many thanks for your time in reviewing our paper and your constructive comments. We have submitted the point-to-point responses. We appreciate if you could let us know whether we have addressed your concerns, and we are happy to answer any further questions.
>
> Best regards,
>
> Authors of paper \#1255

---

### Official Review · Reviewer_Z4Cp · 2025-07-03

**Clarity:** 3
**Significance:** 3
**Originality:** 3
**Rating:** 5
**Confidence:** 3

**Summary:**

The paper introduces a novel approach to flow-based image editing, termed Direct Noise Alignment Editing (DNAEdit). Unlike traditional inversion-based editing techniques, DNAEdit operates by directly refining randomly initialized noise within the Gaussian noise domain, progressively aligning it with a more suitable target noise sample. Additionally, the authors present a technique called Mobile Velocity Guidance (MVG), which enhances the balance between editability and fidelity by calculating the difference between source- and target-conditioned velocity fields in the image domain. Finally, the authors propose a new image editing benchmark that uses long prompts, named DNA-Bench.

Experiments on both PIE-Bench and DNA-Bench, along with qualitative evaluations, demonstrate the competitive performance of the proposed method.

**Questions:**

Overall, the paper is of good quality, presenting an interesting and novel idea supported by strong qualitative and quantitative results. Below, I offer several comments and suggestions to further enhance the quality of the submission:


1. Please add user study results (Weakness 1)

2. Please add KV-edit to comparison  (Weakness 2)

3. Please elaborate how parameter $\eta$ quantitatively influences editing quality  (Weakness 3)

4. Please elaborate on style editing scenarios  (Weakness 4)

**Ethical Concerns:**

["NO or VERY MINOR ethics concerns only"]

**Final Justification:**

Authors have addressed all my concerns. I have checked other reviews and rebuttals and believe that the submission is of the very good quality.

**Limitations:**

yes

**Quality:**

3

**Strengths And Weaknesses:**

## Strengths:

1. The ideas of DNA and MVG are novel and original

2. The paper is well-organized and presented in a clear, comprehensible manner

3. Editing examples and experiments results demonstrate the method's strong performance


## Weaknesses

1. The qualitative results lack user study

2. Table 1 lacks important baseline - KV-edit [1]

3. Equation 10 introduces $\eta$ parameter. Could you please provide quantitative effect of this parameter on method performance?

4. It seems that DNAEdit may have problems with editing tasks where background preservation is not required, such as stylization. Could you please elaborate on that?

[1] Zhu, T., Zhang, S., Shao, J., & Tang, Y. (2025). KV-edit: Training-free image editing for precise background preservation. arXiv preprint arXiv:2502.17363.

---

> ### Author Rebuttal · Authors · 2025-07-30
>
> We sincerely thank this reviewer for the constructive comments and suggestion. We hope our following point-to-point responses can address this reviewer's concerns.
>
> **Weakness 1: The qualitative results lack user study.**
>
> Thanks for the nice suggestion. As suggested by this reviewer, we conducted a user study on the 7 top-performing methods in terms of quantitative metrics. The results are shown in the table below. In this study, we randomly selected 15 images, each edited using the 7 competing methods, and invited 20 participants to evaluate the editing results. We asked the participants to evaluate the edited images by considering the following factors: **the perceptual quality of the edited image, the fidelity to the original image, and whether the editing purpose is achieved**. With these factors in mind, each of the participants selected the best and second-best methods for each image. The statistical results are presented in the table. It can be seen that our DNAEdit-FLUX accounts for 68% of the "best" selections, far exceeding other methods. In the "second-best" selections, our SD-based version also slightly outperforms RF-Solver and FlowEdit-FLUX. These results demonstrate the effectiveness of our method.
>
> | Method           | RF-Solver | FireFlow | InfEdit | FlowEdit-FLUX | FlowEdit-SD | Ours-SD | Ours-FLUX |
> |------------------|-----------|----------|---------|---------------|-------------|---------|-----------|
> | Best (%)         | 9.3       | 3.1      | 4.0     | 6.7           | 0.4         | 8.4     | **68.0**  |
> | Sec-Best (%)     | 22.2      | 13.8     | 5.3     | 20.0          | 1.8         | **27.6**| 9.3       |
>
> **Weakness 2: Table 1 lacks important baseline - KV-edit.**
>
> We thank this reviewer for the suggestion. There is a discrepancy in the settings between KV-Edit and the baseline methods we compared. The baseline methods we need to compare are pure text-guided editing methods, whereas KV-Edit, in addition to text input, requires pre-defined masks as inputs to distinguish editable and non-editable regions. This gives KV-Edit a unique advantage in terms of background preservation metrics such as PSNR for editing tasks. We will include citations and discussions related to KV-Edit in the revision to clarify the differences between methods with varying settings.
>
> **Weakness 3: Equation (10) introduces parameter $\eta$. Could you please provide quantitative effect of this parameter on method performance?**
>
> Thanks for the suggestion. As suggested, we conducted additional ablation studies on the parameter $\eta$. The results are shown in the table below.
>
> | Exp. | $\eta$ | Structure Distance $\downarrow$ | Background Preservation       |                  |                  |                  | CLIP Similarity       |                  |
> |:----:|:------:|:-------------------------------:|:------------------------------:|:----------------:|:----------------:|:----------------:|:---------------------:|:----------------:|
> |      |        |                                 | PSNR $\uparrow$                | LPIPS $\downarrow$ | MSE $\downarrow$ | SSIM $\uparrow$  | Whole $\uparrow$      | Edited $\uparrow$ |
> |  ①   |  1.0   |             33.98               |             21.97              |      149.84      |      95.19       |      79.32       |       **26.40**       |      **23.23**   |
> |  ②   |  0.9   |             24.52               |             23.64              |      118.20      |      67.05       |      83.06       |        26.16          |       22.98      |
> |  ③   |  0.8   |             18.87               |             24.99              |       95.06      |      50.45       |      85.71       |        25.79          |       22.87      |
> |  ④   |  0.7   |           **15.32**             |           **26.06**            |      **79.43**   |     **39.20**    |     **87.58**    |        25.39          |       22.06      |
>
> As can be seen from the above table, when $\eta = 1$, the denoising relies entirely on $v^{\text{tgt}}$. Although it achieves a high CLIP score, the structural distance and background preservation are extremely poor (*e.g.,* the LPIPS metric reaches 149.84).
>
> As $\eta$ decreases to 0.9–0.8, the influence of $v^{\text{mvg}}$ increases, leading to significant improvements in structural consistency and background preservation. The LPIPS metric improves to 95.06, indicating that over-editing is mitigated. Although the overall CLIP score decreases slightly, the CLIP score for the edited region remains relatively stable, only falling from 23.23 to 22.87. This suggests that setting $\eta$ around 0.9–0.8 can guide editing toward the target region while preserving the background.
>
> When $\eta = 0.7$, the improvements in structural and background metrics become limited, and the CLIP score is severely compromised: the CLIP score for the edited region drops to 22.06. An excessively small $\eta$ causes the editing to over-prioritize the original image content, resulting in editing failures.
>
> This ablation study demonstrates the impact of $\eta$ on benchmark performance: an $\eta$ value between 0.9 and 0.8 generally yields better editing results. We will add this study in the revision.
>
> **Weakness 4: It seems that DNAEdit may have problems with editing tasks where background preservation is not required, such as stylization. Could you please elaborate on that?**
>
>
> DNAEdit can also perform well in global editing tasks such as stylization. Although stylization does not require background preservation, information like the layout and colors of the original image still needs to be retained. Therefore, accurately identifying the initial noise is crucial for stylization, and our Direct Noise Alignment can achieve this goal. In addition, the MVG we designed can adaptively calculate the regions that need modification by computing the velocity difference between conditioning on the source prompt and the target prompt. For the target prompt in stylization, it will modify the entire image.
> Due to rebuttal rules, we cannot provide new visual demonstrations of stylization. However, the reviewer can refer to the comparisons in the last two rows on the right side of Figure 10 in the Appendix, where our DNAEdit can achieve the target stylization while preserving the contents of the original image.

---

> ### Author Response · Authors · 2025-08-04
>
> Dear Reviewer Z4Cp,
>
> Many thanks for your time in reviewing our paper and your constructive comments. We have submitted the point-to-point responses. We appreciate if you could let us know whether we have addressed your concerns, and we are happy to answer any further questions.
>
> Best regards,
>
> Authors of paper \#1255

---

> > ### Comment · Reviewer_Z4Cp · 2025-08-07
> >
> > Dear Authors,
> >
> > Thank you for providing the clarifications and additional experiments. Your responses have addressed my concerns, and I am pleased to increase my rating to 5: Accept. I believe that the proposed method is of the very good quality and deserves to be accepted to NeurIPS 2025. I strongly recommend incorporating the key information from your rebuttal into the camera-ready version of the paper, as it will enhance the paper's clarity and overall quality.
> >
> >
> > Best regards,
> >
> > Reviewer Z4Cp

---

> > > ### Author Response · Authors · 2025-08-08
> > >
> > > We sincerely thank this reviewer for the positive feedback and the great support on our work! We will definitely incorporate the contents provided in the rebuttal into the revision. You and the other reviewer’s constructive comments and suggestions indeed helped us a lot to shape this paper stronger.
> > >
> > > Best regards,
> > >
> > > Authors of paper \# 1255

---

### Decision · Program_Chairs · 2025-09-17

**Decision:**

Accept (spotlight)

**Comment:**

This paper presents a new methods for image editing based on T2I rectified flow models.  The approach works by foregoing inversion and instead gradually adjusts the noise sample.  The resulting method demonstrated state-of-the-art results on image editing tasks when compared with multiple recent baselines.

After rebuttal and discussion the reviewers and AC all agreed that the work warranted acceptance.  Authors should be sure to incorporate the additional results and discussions into the final version of the paper.